# STRUCTURED CONSISTENCY LOSS FOR SEMI-SUPERVISED SEMANTIC SEGMENTATION

## ABSTRACT

The consistency loss has played a key role in solving problems in recent studies on semi-supervised learning. Yet extant studies with the consistency loss are limited to its application to classification tasks; extant studies on semi-supervised semantic segmentation rely on pixel-wise classification, which does not reflect the structured nature of characteristics in prediction. We propose a *structured consistency loss* to address this limitation of extant studies. Structured consistency loss promotes consistency in inter-pixel similarity between teacher and student networks. Specifically, collaboration with CutMix optimizes the efficient performance of semi-supervised semantic segmentation with structured consistency loss by reducing computational burden dramatically. The superiority of proposed method is verified with the Cityscapes; The Cityscapes benchmark results with validation and with test data are 81.9 mIoU and 83.84 mIoU respectively. This ranks the *first* place on the pixel-level semantic labeling task of Cityscapes benchmark suite. To the best of our knowledge, we are the first to present the superiority of state-of-the-art semi-supervised learning in semantic segmentation.

## 1 INTRODUCTION

With the deep learning approach of Krizhevsky et al. (2012) introduced in ILSVRC-2012 (Russakovsky et al., 2015), deep learning based image processing has quickly spread among the field. Deep learning has evolved into supervised learning, where the performance of the network is greatly affected by the quantity and quality of labeled data provided. However, preparing labeled data is time-consuming and expensive as it is manually prepared. When it comes to semantic segmentation, it requires pixel-by-pixel annotation making it particularly costly to prepare labeled data for processing. Currently, various techniques such as active learning (Mackowiak et al., 2018), interactive segmentation (Maninis et al., 2018), weakly-supervised leaning (Lee et al., 2019) and more are developed and presented to solve the labeling cost problem in semantic segmentation. In this study, we are aimed at bypassing the labeling cost problem with semi-supervised learning technique.

Semi-supervised learning is a technique to improve network performance with relatively a smaller number of labeled data when a large number of unlabeled data is also available. In reality, a simple build up of unlabeled data can be easily collected from various data sources such as web crawling, vehicle logging and more, and what makes data preparation costly is the manual labelling process to make the collected data useful in learning process. That is, semi-supervised learning technique best suits the data needs of semantic segmentation in real-world research environment. For this reason, our research designed the semi-supervised learning approach suitable for semantic segmentation.

Prior studies apply various versions of semi-supervised learning technique to image classification task and report significant improvement in the performance. Unlike classification, semantic segmentation is a task that performs structured prediction. Specifically, prediction of classification results in a class vector, while semantic segmentation makes predictions per regional location and predicts structural characteristics of regions. We can also learn from prior studies that structural relationship between pixels is important in semantic segmentation; Liu et al. (2019) discussed the distillation of between-pixel relationships and Xie et al. (2018) discussed the neighbor-pixel distillation of features. Together, we pinpoint that it is important to adequately modify the semi-supervised learning to be applied in semantic segmentation.

In this paper, we propose a semi-supervised learning with structured consistency loss in semantic segmentation. Structured consistency loss reflects the structural relationship between pixels in semantic segmentation. It regularizes the inter-pixel relationships consistent, thereby allowing the network to learn more powerful generalization capabilities. We propose a method incorporating CutMix (Yun et al., 2019) in data augmentation, achieving a significantly lower GPU memory utilization by limiting the observation region at each CutMix box. This allows us to process the final prediction map at the size of the original image; we no longer must reduce the size of the final prediction map to lower the GPU memory utilization, unlike it was with pair-wise knowledge distillation which requires feature distillation with the lowest resolution.

Our method achieved the first place with mIOU 83.84 in the Cityscapes benchmark pixel-level semantic labeling task (Cordts et al., 2016). This is the first study showing that the semi-supervised learning achieved the state-of-the-art performance in semantic segmentation using the Cityscapes benchmark. Note that our semi-supervised learning technique could be used in parallel with other researches for further improvements since our contribution is not in network architecture, but in learning techniques.

## 2 RELATED WORKS

**Semantic Segmentation**. The early period of semantic segmentation approaches were mostly based on Fully Convolutional Network (FCN) (Sermanet et al., 2013; Long et al., 2015). Improving earlier segmentation models, the loss of spatial information is mediated with the use of encoder-decoder architecture (Noh et al., 2015; Ronneberger et al., 2015; Badrinarayanan et al., 2017) or dilated convolution (a.k.a. Atrous Convolution) expanding the receptive field (Chen et al., 2014; Yu & Koltun, 2015). Improving the localization performance, (Chen et al., 2017) applied the Atrous Spatial Pyramid Pooling (ASPP) in semantic segmentation and PSPNet (Zhao et al., 2017) proposed a feature pyramid pooling module to gather global contextual information around the image object or stuff.

More recently, Chen et al. (2018) suggested well-organized architecture with combining encoder-decoder architecture and dilated convolution. Many subsequent methods that achieved state-of-the-art performance have followed this structure (Takikawa et al., 2019; Zhuang et al., 2018; Li et al., 2019). Zhu et al. (2019) adopted DeeplabV3Plus (Chen et al., 2018) with WideResNet38 (Zagoruyko & Komodakis, 2016) as the backbone network. The inspiring results were generated due to labelling of video image using temporal information, not by architectural improvements. Moreover, boundary label relaxation also eased the strictly labeling in boundary issue and class uniform sampling dealt the class imbalance problem. To make our contribution clearly visible, we employed those architecture and methods except for video label generation .

**Semi-Supervised learning**. In recent years, semi-supervised method has become a very prominent theme, but a lot of researches were limited to classification task in earlier times. They used a loss function computed on unlabeled data and encouraged the model to generalize better to unseen data in same domain. Grandvalet & Bengio (2005) proposed the entropy minimization loss showing that decision boundary tends to lie on low density region of class distribution. Consistency loss which encourages the model to produce the same output distribution when its inputs are perturbed is suggested in Laine & Aila (2016). Consistency regularization loss played a breakthrough role which facilitated researches in this field.

To employ consistency loss efficiently, Tarvainen & Valpola (2017) proposed teacher network with exponential-moving-average (EMA) network and generated better guessed label. More recently, MixMatch (Berthelot et al., 2019) combined entropy minimization, consistency loss, and MixUp (Zhang et al., 2017) regularization for the better generalization. Furthermore, result of Xie et al. (2019) nearly matched the performance of models trained on the full sets of CIFAR-10 with only using 10 % of this dataset, thanks to sophisticated augmentation method with realistic noise.

**Structured Prediction**. Semantic segmentation has a nature of structured prediction; prediction result has a equal shape of input image and prediction vector of each pixel has a strong correlation with each other, especially close one. There were some researches reflected this characteristics by a

loss term, or architectural perspective. In Xie et al. (2018), local pair-wise (8-neighbors) distillation was used to make efficient feature distillation. They transferred the knowledge of zero-order, and first-order by consistency map. Liu et al. (2019) showed impressive progress of feature distillation using pair-wise distillation. They calculated total inter-pixel similarities in a specific feature map and distilled the feature of teacher to student by similarities.

The knowledge distillation and consistency loss differ in the underlying philosophy. The purpose of knowledge distillation is to enable an efficient network to learn powerful network predictions or feature extraction capabilities for the same image, while the goal of consistency loss is to enhance robustness against disturbances by forcing the same prediction. However, the implementation detail of both methods are same. That is, using the distance function as the cost function. In this paper, we employed the knowledge distillation technique introduced in Liu et al. (2019) to our structured consistency loss.

Most recently, consistency loss with CutMix (Yun et al., 2019) for semantic segmentation was introduced in French et al. (2019). They investigated the difference between classification and semantic segmentation in terms of low density region; it lies in boundary of class distribution in classification, while it is distributed in object boundary pixels of each image in semantic segmentation. For this reason, they selected CutMix instead of MixUp to conserve the local-structural identity of image in CutMix box. We also used CutMix augmentation to give different perturbed images to teacher and student while maintaining the structural information.

## 3 METHODOLOGY

Since semantic segmentation can be considered as a classification problem performed pixel by pixel, it is reasonable to try to apply semantic segmentation in the same way to consistency loss, widely used in semi-supervised classification. Indeed, several studies have made improvements in this methodology. However, semantic segmentation has a characteristic of structured prediction unlike general classification. For this reason, if the existing consistency loss is used as it is, it is difficult to expect high performance improvement while ignoring the nature of the structured prediction of semantic segmentation.

Therefore, structured consistency loss is used to encourage network to predict properly focusing on inter-pixel relationship in local patch using cosine similarity. Local patches are implemented with random CutMix augmentation scheme. Looking at too narrow range of receptive field can be ineffective in identifying pixel-to-pixel relationships, while looking at too wide range increases computational complexity and hinder network's learning. Therefore, precise tuning of the CutMix box size is essential. Experimental details would be explained in section 4.1.

### 3.1 ARCHITECTURE

Our main architecture is divided into two parts, supervised training with labeled image and unsupervised training with unlabeled image. These images are used to compose mini-batch in each step in order to calculate labeled and unlabeled loss simultaneously. Formally, the combined loss $\mathcal{L}_{tot}$ for semi-supervised learning is computed as:

$$\mathcal{L}_{tot} = \mathcal{L}_x + \mathcal{L}_u \tag{1}$$

$$\mathcal{L}_x = -log \sum_{C \in \mathcal{N}_w} P(C) \tag{2}$$

Where $\mathcal{L}_x$ is general cross entropy loss with boundary label relaxation for semantic segmentation with labeled image $x$. $\mathcal{L}_u$ is a loss with unlabeled image $u$. In equation (2), $\mathcal{L}_x$ is composed of $P()$ with softmax probability of each class where $\mathcal{N}_w$ is the set of classes within a $w$ by $w$ window boundary region, respectively. Note that $\mathcal{L}_x$ reduces to the standard cross entropy loss with one-hot label when $w$ equals one.

**CutMix**. The CutMix augmentation was originally designed for classification task. It mixes samples by cutting rectangular patch and pasting from one sample into another. Following as done

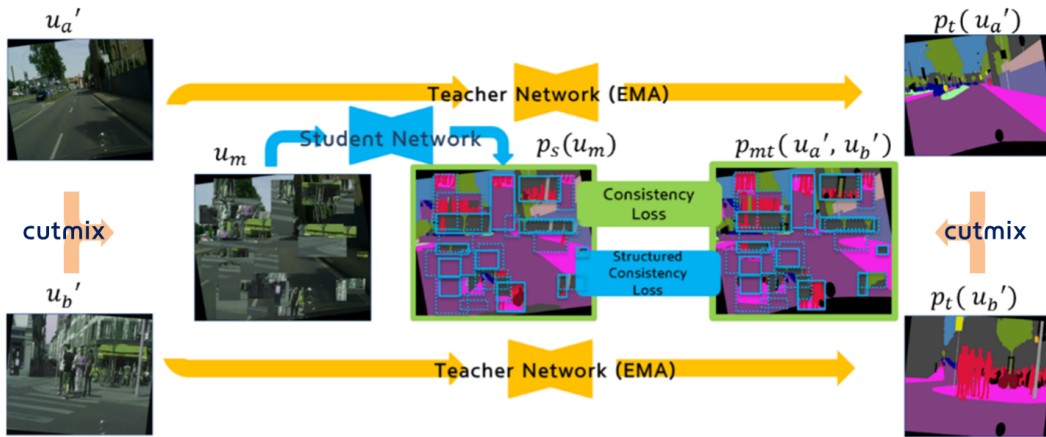

Figure 1: Main architecture describing semi-supervised approach with CutMix augmentation for semantic segmentation. Two losses, consistency and structured consistency, are calculated from prediction of teacher and student networks each.

by Liu et al. (2019), we use it for semantic segmentation by generating $N$ patches with random size and position. The total area of patches would be approximately half of the image dimension $H, W$ to make balance between two images $u$ and $x$.

**Semi-supervised**. In Figure 1, our main architecture of semi-supervised approach is described. First, we generate $\acute{u_a}$ and $\acute{u_b}$ by applying random augmentation used in Zhu et al. (2019) except CutMix. And then create CutMix image $u_m$ from the augmented images. The teacher network takes the images as input and generates prediction results ($p_t(\acute{u_a})$, $p_t(\acute{u_b})$), and also generates CutMix image $p_{mt}(\acute{u_a}, \acute{u_b})$ from the two predictions. Student network use the result as a *guessed label* introduced by Berthelot et al. (2019). The student network gets the CutMix image as input during training and produces prediction $p_s(u_m)$. In the end, we calculate two consistency losses using the prediction of student and teacher networks each. Note that teacher and student network have same network architecture. The weights of teacher is not from its own weights using gradient update, but from the student weights by EMA manipulation. These losses give the student network ability to predict similar result as the teacher networks by only looking at local patches where teacher network receives the overall image information.

Unlabeled loss term is separated into two parts: consistency loss and structured consistency loss. Consistency loss is a conventional loss used for classification task and performed pixel-wise manner while structured consistency loss is a novel approach to figure out inter-pixel relationship within local patch. Formula of unlabeled loss is as follows:

$$\mathcal{L}_u = \lambda_c \mathcal{L}_c + \lambda_{sc} \mathcal{L}_{sc} \tag{3}$$

$$\mathcal{L}_c = \frac{1}{H \times W} \sum_{i \in \mathbb{T}} ||p_s(u_{m,i}) - p_{mt}(\acute{u}_{a,i}, \acute{u}_{b,i})||^2 \tag{4}$$

Where $\mathcal{L}_c$ is conventional consistency loss which is average of pixel-wise squared L2 loss used in classification task. We will deep dive into $\mathcal{L}_{sc}$ in next section. $\lambda_c, \lambda_{sc}$ are weight for each loss term. $\mathbb{T} = \{1, 2, ..., H \times W\}$ denotes a set of all the pixel indices in the image. $p_s(u_{m,i})$ represents prediction of student network for $i$th pixel of CutMix unlabeled image $u_m$. $p_{mt}(\acute{u}_{a,i}, \acute{u}_{b,i})$ represents CutMix prediction of teacher network for $i$th pixel for unlabeled augmented images $\acute{u_a}, \acute{u_b}$.

## 3.2 STRUCTURED CONSISTENCY LOSS

As explained in section 2, we used pair-wise knowledge distillation technique suggested by Liu et al. (2019) into structured consistency loss can be written as:

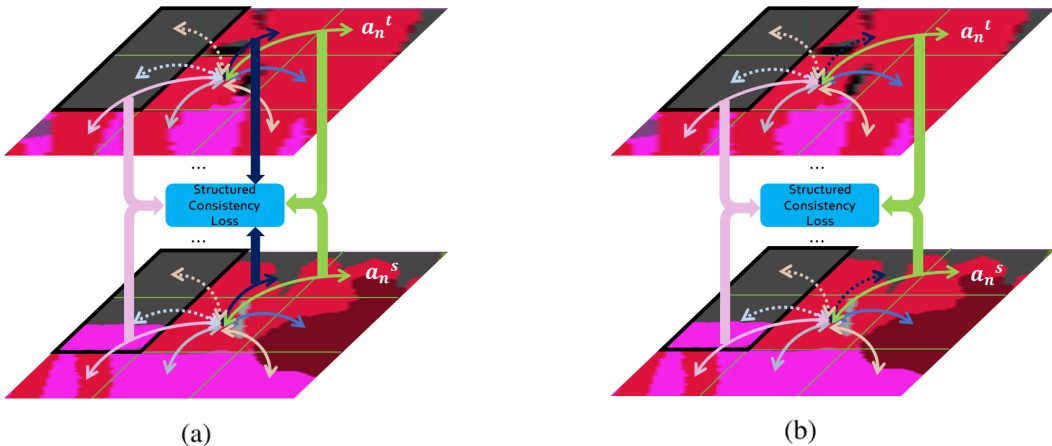

(a)                                                                    (b)

Figure 2: Structured consistency loss calculation procedure is described visually. CutMix box size of 3 by 3 is used for simplicity. Note that the each pair of cosine similarity $(a_n^s, a_n^t)$ with same color are used to calculate the loss. (a) Masking the covered region (denoted as black box) by another CutMix box since it doesn't include uncovered box related information. (b) Random elimination of cosine similarities when the number of similarities exceed $N_{pair}$.

$$a_{ij} = \boldsymbol{p}_i^T \boldsymbol{p}_j / (||\boldsymbol{p}_i||||\boldsymbol{p}_j||) \tag{5}$$

$$\mathcal{L}_{sc} = \frac{1}{(H \times W)^2} \sum_{i \in \mathbb{T}} \sum_{j \in \mathbb{T}} ||a_{ij}^s - a_{ij}^t||^2 \tag{6}$$

Where $a_{ij}^t$ and $a_{ij}^s$ denote the similarity between the $i$th pixel and the $j$th pixel produced from the teacher and student network each and $\boldsymbol{p}_i$ represents prediction vector of $i$th pixel.

In the previous work, prediction vector ($\boldsymbol{p}$) was calculated at the feature stage rather than the prediction stage due to feature distillation effects. On the contrary, structured consistency loss must be calculated at the image domain because the goal is to get the same prediction from the perturbed image. In order to do that, there are issues to overcome to calculate structured consistency loss in image domain; The calculation of structured loss consumes a lot of GPU memory usage since the complexity is $(H \times W)^2$ , not feasible in image domain with the same size as the input.

Nevertheless, because our main architecture uses CutMix augmentation, calculating the structured consistency loss inside the CutMix box reduces computing complexity approximately $N$ times. To be specific, CutMix augmentation allows limitation of calculation into the CutMix box, thereby enabling efficient GPU memory usage. Moreover, by using CutMix, the network can learn the ability to make accurate predictions with limited local information which predicts similar to the prediction with global area. The structured consistency loss with CutMix is described as below:

$$a_{n,ij} = \boldsymbol{p}_{n,i}^T \boldsymbol{p}_{n,j} / (||\boldsymbol{p}_{n,i}||||\boldsymbol{p}_{n,j}||) \tag{7}$$

$$\mathcal{L}_{sc} = \sum_{n=1}^{N} \frac{1}{|\mathbb{T}_n|^2} \sum_{i \in \mathbb{T}_n} \sum_{j \in \mathbb{T}_n} ||a_{n,ij}^s - a_{n,ij}^t||^2 \tag{8}$$

Where $\mathbb{T}_n = \{1, 2, 3, ..., H_n \times W_n\}$ denotes a set of all the pixel indices in $n$th CutMix box except for the region covered by other CutMix box. $a_{n,ij}$ represents the similarity between the $i$th pixel and the $j$th pixel in $n$th CutMix box and $N$ is the number of CutMix boxes.

To achieve state-of-the-art performance, it is necessary to revise the loss term because input image with high resolution and complex network architecture cause the out of memory problem. In order to handle this issue, the number of CutMix boxes for calculating structured consistency loss and the maximum number of pairs for calculating losses in each box are restricted to $N_{box}$ and $N_{pair}$, respectively, to make algorithm realizable. To be specific, we didn't use CutMix box with dotted

line borders while only considered $N_{box}$ CutMix box with straight line for structured consistency loss in Figure 1. In Figure 2(a), we eliminated the region covered by another CutMix box, and also randomly dropped out similarity vectors excessing over $N_{pair}$ in Figure 2(b). Finally, we derived the feasible structured consistency loss as follows:

$$\mathbb{T}^s_{n,drop} = Drop\langle\{(i,j)|\forall i \in \mathbb{T}_n, \forall j \in \mathbb{T}_n\}, N_{pair}\rangle \tag{9}$$

$$\mathcal{L}_{sc} = \sum_{n=N-N_{box}+1}^{N} \frac{1}{|\mathbb{T}^s_{n,drop}|} \sum_{(i,j)\in\mathbb{T}^s_{n,drop}} ||a^s_{n,ij} - a^t_{n,ij}||^2 \tag{10}$$

The customized function $Drop\langle \mathbb{X}, n\rangle$ randomly clips $\mathbb{X}$ to $n$ when exceeding $n$. In the process of attaching the CutMix box, box attached earlier is likely to be covered by the box attached behind, so the loss is calculated using posterior $N_{box}$ of boxes. In addition, the loss is not calculated through masking on the area covered by another box. By doing this, only those pixels that are structural-relevant within the CutMix box can learn about the relationship to each other.

## 4 EXPERIMENTS

### 4.1 IMPLEMENTATION DETAILS

**Network Structures**. We adopt a state-of-the-art segmentation architecture from Zhu et al. (2019), Deeplabv3+ with WideResNet38. *Output stride* of encoder is set to 8 and that of the low level feature transferred to decoder is set to 2. The teacher network used to generate guessed label is made from EMA model (Tarvainen & Valpola, 2017). Inference with test and validation sets are also done with EMA model. The parameter value of EMA weight is set to 0.999 and it is averaged out in every training steps.

**Training Procedure**. We use a SGD optimizer with polynomial learning rate policy. Following the setup of Zhu et al. (2019) the initial learning rate is 0.002, power set to 1.0, weight decay of 0.001 and momentum of 0.9. We use synchronized batch normalization with a batch size of 2 per GPU, one for labeled data and the other for unlabeled one, on 8 V100 GPUs. The training epoch is set to 175 based on the labeled data. For unlabeled data, we adopt CutMix augmentation additionally. From the empirical results, we set the hyper-parameters $N$, $N_{box}$, $N_{pair}$ $\lambda_c$, $\lambda_{sc}$ to 32, 16, 9000, 20, 3 respectively. We use training skills, class uniform sampling and boundary label relaxation, to get strong baseline network.

**Cityscapes**. Cityscapes is a widely accepted validation medium among recent studies, thanks to its high-quality labeled image database. Cityscapes offers pixel-level annotations of 5,000 images and coarse annotations of 20K images. The reliable pixel-level annotations are also split into three subsets for training, validation and test, each consisting of 2,975, 500 and 1,525 images respectively, and we used these training, validation and test set to gauge the performance of our methodology presented in this paper. We also use coarsely annotated images for class uniform sampling to overcome the imbalance between classes. Those images with coarse annotations are also used, ignoring the annotations associated with, as unlabeled images in unlabeled training phase for greater learning exposure.

### 4.2 RESULTS

The experimental results from our semi-supervised learning technique with structured consistency loss are summarized in Table 1. Our experiments were conducted on Cityscapes test images with multi-scale strategy (0.5, 1.0 and 2.0), horizontal-flip, and overlapping-tile methods. Our performance result ranks the first in the Cityscapes benchmark, even after including the results of unpublished studies. We also want to emphasize that the result with only training set exhibits the highest performance result when compared with performance results of published methods. Graphical results on the Cityscapes test set are presented in Figure 3.

Table 1: Comparison of Per-class mIoU results of recent methods on Cityscapes with our state-of-the-art method. Both published and unpublished methods are included. Our results exhibit the best overall performance. Note that the results for "Ours" is obtained from training data only, and "Ours+" is obtained from training and validation data.

| Method | road | s.walk | build. | wall | fence | pole | t-light | t-sign | veg | terrain | sky | person | rider | car | truck | bus | train | motor | bike | mIoU |
|---|---|---|---|---|---|---|---|---|---|---|---|---|---|---|---|---|---|---|---|---|
| Gated-SCNN | 98.7 | 87.4 | 94.2 | 61.9 | 64.6 | 72.9 | **79.6** | 82.5 | **94.3** | 73.3 | **96.2** | 88.3 | 74.2 | **96.6** | 77.2 | 90.2 | 87.7 | 72.6 | 79.4 | 82.8 |
| DRN-CRL | 98.8 | 87.7 | 94.0 | **65.0** | 64.2 | 70.2 | 77.4 | 81.6 | 93.9 | 73.5 | 95.8 | 88.0 | 74.9 | 96.5 | **80.8** | 92.1 | 88.5 | 72.1 | 78.8 | 82.8 |
| GALD-Net | 98.8 | 87.7 | 94.2 | 65.0 | **66.7** | **73.1** | 79.3 | 82.4 | 94.2 | 72.9 | 96.0 | **88.4** | **76.2** | 96.5 | 79.8 | 89.6 | 87.7 | **74.1** | **79.9** | 83.3 |
| Video Propagation | 98.8 | 87.8 | 94.2 | 64.0 | 65.0 | 72.4 | 79.0 | 82.8 | 94.2 | 74.0 | 96.1 | 88.2 | 75.4 | 96.5 | 78.8 | **94.0** | **91.6** | 73.7 | 79.0 | 83.5 |
| Ours | **98.8** | **88.2** | **94.3** | 64.5 | 65.3 | 72.6 | 79.3 | **82.9** | 94.2 | **74.2** | 96.1 | 88.4 | 75.6 | 96.6 | 79.5 | 93.7 | 91.2 | 73.7 | 79.4 | **83.6** |
| Tencent AI Lab | 98.6 | 86.9 | 94.1 | 63.5 | 63.0 | 70.7 | 77.7 | 80.2 | 94.0 | 73.1 | 95.9 | 87.8 | 74.5 | 96.3 | 82.8 | 94.3 | 90.4 | 74.0 | 77.5 | 82.9 |
| GGCF | 98.8 | 87.8 | 94.1 | 66.0 | 66.1 | 71.1 | 78.4 | 82.2 | 94.1 | **74.5** | 95.8 | 88.0 | 74.0 | 96.5 | 79.9 | 92.4 | 90.8 | 71.8 | 78.4 | 83.2 |
| iFLYTEK-CV | 98.7 | 87.1 | 94.1 | 64.4 | 65.4 | 71.2 | 77.9 | 82.2 | 94.0 | 73.5 | 96.0 | 88.3 | 75.7 | 96.5 | **83.3** | **94.7** | **92.4** | 74.3 | 79.0 | 83.6 |
| openseg-group | 98.8 | 88.3 | 94.3 | **66.9** | **66.7** | **73.3** | **80.2** | **83.0** | 94.2 | 74.1 | 96.0 | 88.5 | 75.8 | 96.5 | 78.5 | 91.8 | 90.2 | 73.4 | 79.3 | 83.7 |
| Ours+ | **98.9** | **88.4** | **94.3** | 65.2 | 65.9 | 72.8 | 79.5 | 83.0 | **94.3** | 74.3 | **96.1** | **88.6** | **75.9** | **96.6** | 79.3 | 93.8 | 91.5 | **74.8** | **79.7** | **83.8** |

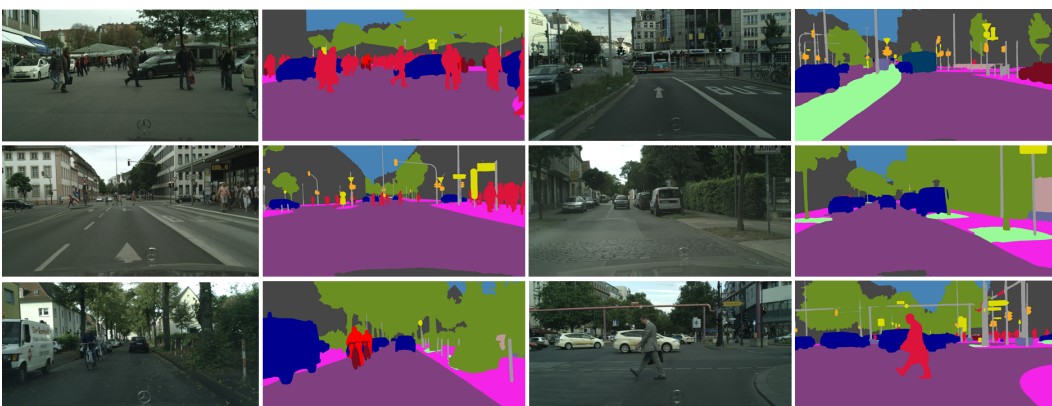

Figure 3: Qualitative results of our method on the Cityscapes test set. Pictures depict predicted segmentation masks.

We believe the observed improvement in performance is attributed to (a) the use of coarsely labeled images in unlabeled training phase and (b) the effectiveness of structured consistency loss added to semi-supervised learning in semantic segmentation. In recent years, many researchers have attempted to achieve the close performance obtained from learning with the entire labeled data set by using a combination of only a small portion of the labeled data set and a large unlabeled data set. Yet the use of coarsely labeled data has been limited in extant studies; they are only used in the early training stage, or sporadically used to aid the segmentation of images with rare classes.

We believe that our approach reflects the real-world environment where the available training resources are mostly unlabeled; the combination of a small labeled data set and a large unlabeled data set to achieve the performance equivalent to using a large labeled data set is the way we all want to go. We are a step closer to this goal with the structured consistency loss applied to semi-supervised learning in semantic segmentation.

## 4.3 ABLATION STUDY

**Validation with Baseline**. In this section, we conducted additional experiments to demonstrate the superiority and effectiveness of our proposed method in semi-supervised learning. We first perform a supervisory baseline, and the result is summarized in the first row of Table 2. The baseline result exhibits a reasonable performance, as a decent amount of recent techniques are already included; pre-training with Mapillary, boundary label relaxation, and class uniform sampling are included in the baseline. By adding consistency loss to the baseline, the mIoU increases to 81.55% from 81.18%, showing the practical implication of the semi-supervised learning using the CutMix in semantic segmentation. mIoU increases further with addition of structured consistency loss to 81.90%, reflecting the significance of our proposed method.

The qualitative results have been visually provided in Figure 4, allowing us to gauge the qualitative performance. Thanks to inter-pixel consistency processed with the structured consistency loss, our proposed method exhibits less errors in small and visually easy-to-confuse region(s).

**Exponential-moving-average (EMA) Application.** Networks with EMA are well-known for their great generalization ability (Tarvainen & Valpola, 2017; Berthelot et al., 2019). Hence it must be beneficial to apply EMA to both training (teacher network) and inference (validation or test). Nevertheless, not all recent studies apply EMA to the weight of teacher network, but use the same weight as that of student network. Therefore, we conduct an experiment to examine the performance variability depending on application of EMA. The experiment result is sumamrized in Table 3. When the EMA weight is used in the teacher network, mIoU increases 0.135% on average, while the use of EMA weight in validation seems to have no significant effect on mIoU. Since the use of EMA weight in teacher network and validation yields the greatest mIoU, we proceed to use the EMA weight in both.

Table 2: Comparison of mIoU results of supervised-only (baseline), with consistency loss, with consistency loss and structured consistency loss on Cityscapes validation images.

Table 3: Comparison of mIoU depending on whether exponential-moving-average (EMA) is applied to both/either teacher network and/or validation on Cityscapes image.

| Method | mIoU (%) | Gain (%) |
|---|---|---|
| Baseline | 81.17 | 0.0 |
| + Consistency loss | 81.55 | +0.38 |
| + Structured Consistency loss | 81.90 | +0.35 |

| EMA | | mIoU (%) |
|---|---|---|
| Teacher | Validation | |
| X | X | 81.77 |
| X | O | 81.74 |
| O | X | 81.88 |
| O | O | 81.90 |

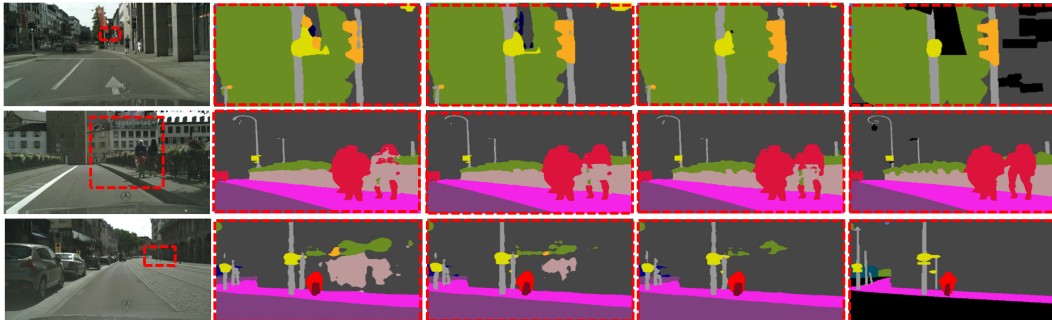

Figure 4: Qualitative results on the validation data of Cityscapes. From the left to the right are original images, supervised-only, addition of consistency loss, addition of structured consistency loss, and ground truth.

## 5 CONCLUSION

In this paper, we propose the structured consistency loss in semi-supervised learning for semantic segmentation. With CutMix augmentation, the structured consistency loss fully exploits the relationship among local regions and enhance the network generalization. This technique could also be simultaneously applied to the general network. Our proposed method achieves the state-of-the-art premier performance in the Cityscapes benchmark suite, not only ranking the first place only among publication results, but also among all results including unpublished results. Therefore, we are confident that the semi-supervised learning is also highly effective to solve practical real world problems under data-insufficiency, when accompanied with CutMix augmentation as well as the structured consistency loss. Despite of superior performance exhibited by our method, the random sampling using $N_{pair}, N_{box}$ in the proposed method still has a room for optimization. We leave this as a future work.

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
