# OpenReview forum: "Structured consistency loss for semi-supervised semantic segmentation"
_ICLR.cc/2020/Conference — Reject_

### Official Review · AnonReviewer1 · 2019-10-23
**Official Blind Review #1**

**Rating:** 1

**Review:**

- This paper introduces a structured consistency loss for semantic segmentation under the semi-supervised learning paradigm. Specifically, authors propose to include this term to force consistency between the predictions of two networks (a teacher and a student). To evaluate their method, authors employ images from the cityscapes dataset. Results show that the proposed method achieves competitive results compared to state-of-the-art architectures.
- Contributions are unclear. The only contribution seems to be the ‘first study that employs semi-supervised learning and achieves state-of-the-art performance in semantic segmentation using the Cityscapes’. First, this contribution is insufficient to be considered as a technical contribution. And second, I feel authors mislead the reader with this assertion.
- The paper is poorly written with many senseless sentences and weird word compositions (e.g., ‘our research designed the semi-supervised learning approach suitable for semantic segmentation’ or ‘In this study, we are aimed at bypassing the labeling cost problem with semi-supervised learning technique.’ or ‘It regularizes the inter-pixel relationships consistent, thereby allowing the network to learn more powerful generalization capabilities’ just to give few examples). This makes the reading of this work particularly difficult, as well as the understanding on some parts.
- Authors fail to compile relevant works, particularly in semi-supervised segmentation, leading to a very weak related work section. Particularly in semi-supervised segmentation, more than one hundred papers have been published since 2015 only in major conferences on vision (CVPR,ICCV,ECCV) and learning (ICLR, ICML, NeurIPS). None of them, however, are listed in the related work section. Authors should significantly improve this section.
- Which is the difference with French et al. (2019) and Liu et al. (2019)? Both seem to be using CutMix for semantic segmentation.
- Where the labeled images are used? Authors show that labeled images and their corresponding ground truth masks are employed in eq 1) and 2). Nevertheless, it is never explained in which moment these are employed. Are they used to pre-traine the student and teacher networks? Are they mixed with unsupervised images during training? Please clarify.
- Please add reference for the cross entropy loss with boundary label relaxation.
- Furthermore, this paper contains many grammatical errors.
- Even though the idea is interesting, I am inclined to reject this paper because there are many unclear details, the related work is insufficient and the paper needs a thorough proof-reading in english.

**Experience Assessment:**

I have published in this field for several years.

**Review Assessment: Checking Correctness Of Derivations And Theory:**

I carefully checked the derivations and theory.

**Review Assessment: Checking Correctness Of Experiments:**

I carefully checked the experiments.

**Review Assessment: Thoroughness In Paper Reading:**

I read the paper thoroughly.

---

### Official Review · AnonReviewer3 · 2019-10-23
**Official Blind Review #3**

**Rating:** 1

**Review:**

Summary:
- key problem: semi-supervised semantic segmentation;
- contributions: 1) combine the CutMix data augmentation of Yun et al 2019 with the standard consistency loss of  and the  structured consistency loss of Liu et al 2019, 2) state of the art results on Cityscapes.

Recommendation: Reject

Key reason 1: limited novelty.
- What is the major difference with French et al 2019? Is it just the addition of the structured consistency loss of Liu et al? Figure 1 is almost identical to Figure 3 in French et al 2019.
- What is the major novelty / insight besides the direct combination of the aforementioned papers and others like MixMatch (Berthelot et al 2019)?
- Is the main contribution the faster computation of the structured consistency loss by limiting it to the CutMix boxes?

Key reason 2: limited experimental validation.
- The performance improvements are all <0.4%: are they statistically significant? What is the variance w.r.t. the random seed (important as the method is heavily relying on sampling random transformations)?
- I would suggest conducting experiments on more than just one dataset, e.g., by adding COCO and/or Mapillary Vistas.
- What is the performance of the approach when using only a subset of the labels (e.g., 10%, 25%, 50%) vs the full Cityscapes training set?

Key reason 3: writing could be significantly improved.
- There are numerous grammatical mistakes and unclear sentences, e.g., "It regularizes the inter-pixel relationships consistent", "a lot of researches were".
- Figures 2a and 2b are identical.




**Experience Assessment:**

I have published one or two papers in this area.

**Review Assessment: Checking Correctness Of Derivations And Theory:**

I assessed the sensibility of the derivations and theory.

**Review Assessment: Checking Correctness Of Experiments:**

I assessed the sensibility of the experiments.

**Review Assessment: Thoroughness In Paper Reading:**

I read the paper at least twice and used my best judgement in assessing the paper.

---

### Decision · Program_Chairs · 2019-12-19

**Decision:**

Reject

**Comment:**

This submission proposes to combine the CutMix data augmentation of Yun et al 2019 with the standard consistency loss of  and the  structured consistency loss of Liu et al 2019 and applies the resulting approach to the Cityscapes dataset.  The reviewers were unanimous that the paper is not suitable for publication at ICLR due to a lack of novelty in the method.  No rebuttal was provided.